# Thin-Film Processing of Polypropylene and Polystyrene Sheets by a Continuous Wave CO_2_ Laser with the Cu Cooling Base

**DOI:** 10.3390/polym13091448

**Published:** 2021-04-30

**Authors:** Nobukazu Kameyama, Hiroki Yoshida, Hitoshi Fukagawa, Kotaro Yamada, Mitsutaka Fukuda

**Affiliations:** 1Department of Electrical, Electronic and Computer Engineering, Faculty of Engineering, Gifu University, 1-1 Yanagito, Gifu City 501-1193, Japan; hiroki@gifu-u.ac.jp; 2Institute for Advanced Technology, Heiwadori, Seki City 501-3874, Japan; fukagawa@sengiken.com; 3ATHEN KOGYO CO.LTD, Shimouchi, Seki City 501-3217, Japan; koyamada@athena-kogyo.co.jp (K.Y.); mfukuta@athena-kogyo.co.jp (M.F.)

**Keywords:** laser processing, polypropylene, polystyrene, CO_2_ laser, thin-film processing

## Abstract

Carbon dioxide (CO_2_) laser is widely used in commercial and industrial fields to process various materials including polymers, most of which have high absorptivity in infrared spectrum. Thin-film processing by the continuous wave (CW) laser is difficult since polymers are deformed and damaged by the residual heat. We developed the new method to make polypropylene (PP) and polystyrene (PS) sheets thin. The sheets are pressed to a Cu base by extracting air between the sheets and the base during laser processing. It realizes to cut the sheets to around 50 µm thick with less heat effects on the backside which are inevitable for thermal processing using the CW laser. It is considered that the boundary between the sheets and the base is in thermal equilibrium and the base prevents the sheets from deforming to support the backside. The method is applicable to practical use since it does not need any complex controls and is easy to install to an existing equipment with a minor change of the stage.

## 1. Introduction

Since carbon dioxide (CO_2_) laser was developed by C.K.N. Patel in 1964 [1], it has been widely utilized in many fields including industry and medical. Laser processing is one of the most useful application of CO_2_ laser in industry because it is easy to obtain high power beams with reasonable cost and has many advantages of, for example, less consumables like drills and cutters, and clear kerf with no burrs, compared with mechanical processing. There are many classifications of laser processing: cutting, drilling, welding, marking, annealing, and so on. CO_2_ laser is applicable to process various materials including steels, woods, polymers, etc. [2]. Especially since most polymers are opaque in infrared spectrum, they are effectively processed by CO_2_ laser.

Lasers are mainly classified into two groups: continuous wave (CW) laser and pulse laser. CW lasers can process materials fast and are suitable for mass production as long as heat-affected zone (HAZ) is ignorable, while pulse lasers enable fine processing without HAZ though the processing speed is slow [3]. In industrial fields requiring high speed processing, CW lasers are selected because of low cost of polymers.

Laser cutting is one of the main methods of laser processing as well as welding and drilling [4,5]. The mechanism of the cutting process is segregated into three types: melt sheering, vaporization, and chemical degradation. The cutting mechanism of thermoplastics including polypropylene (PP) and polystyrene (PS) is categorized to melt sheering [2]. In the cutting procedures, the laser beam melts the material and the molten material is ejected by an assistance gas jet acting coaxially with the beam.

Since the thermoplastics are softened by heating, they are inevitable to deform by residual heat after laser scribing, which makes it difficult to reduce the thickness of the sheets with less heat influence. The processing to make polymer sheets thin by laser beams, therefore, is not popular, unlike marking, which scribes shallowly the surface of a workpiece, and engraving, which processes a material that has enough thickness compared with the depth of the groove [6]. There are many methods to make thin polymer films: Langmuir–Blodgett (LB) method [7], electrochemical polymerization [8], chemical vapor deposition (CVD) [9], epitaxy [10], etc. The methods are suitable to make plane films. Lithography, etching, electro discharge machining (EDM), precision machining, and blasting can cut grooves on polymer sheets [11,12,13,14,15,16]. These methods are applied to fabricate microreactors used for polymerase chain reaction (PCR) [17], which have microchannels whose width is several hundred micrometers. Some polymers are available for the material of the microreactors since the polymers are cheap and easily processed even though the chemical resistance and operating temperature range are inferior to silicon and glass [18,19]. Since PCR needs precise temperature control [20], the thickness of the microreactor of which material has low heat conductivity like the polymers should be thin. Thin-film processing by the laser beam has possibility for fabrication of the polymer microreactors because of the advantages of the fast beam scanning, the easy changes of the fabricated shape, and the simple operations of the laser.

In this study, polypropylene (PP) and polystyrene (PS) are selected from among several polymers. Their products are shown in every area of life and are essential for each field because of their usefulness [21,22]. Recently, pollution of microplastic is one of the most important problems in the world [23,24,25]. They are ones of the main components of microplastics [26]. Attempts of reduction of polymer usage have to be continued constantly in all fields. Thin-film processing can be applied for commercial products that have flexible shape and are deformed readily, leading to reducing polymer usage and facilitating garbage collection.

We developed the new method to make PP and PS sheets partially thin with a CW CO_2_ laser by attaching a Cu cooling base to the backside of the sheets during laser processing. Other metals which are aluminum, brass, and steel are confirmed to be available to the cooling base.

## 2. Materials and Methods

The experiments are performed with a CW CO_2_ laser (LP-431U, Panasonic, Kadoma, Japan) that the wavelength is 10.6 µm, the beam is controlled with Galvo mirror system, the spot size is around 110 µm, and the focal length is 111 mm. In this study, the power *P* is fixed to 30 W during the experiments and the laser scanning speed *S* is controlled from 5 to 1500 mm/s. 

Polymer sheets used in this study are PP and PS that both are transparent and around 300 µm in thickness. Absorptivities *A* of PP and PS at the wavelength of 10.6 µm measured by Fourier transform infrared spectrometer (FT-IR) (IRPresige-21, SHIMADZU, Kyoto, Japan) are 60% and 91%, respectively. The absorption coefficient α is calculated by using the equation, $\alpha =-\ln\left(1-A\right)/d$, where *d* is thickness of the sample. Those of PP and PS are 31 cm^−1^ and 81 cm^−1^, respectively. The thermal characteristics of PP measured by differential scanning calorimetry (DSC), simultaneous DSC and thermogravimetric analysis (TGA) (SDT), and thermal–mechanical analysis (TMA) are shown in Figure 1. All measurements are performed under nitrogen atmosphere. The melting and sublimating temperatures are around 160 °C and 460 °C, respectively. The specific heat of PP at 25 °C is 1.4 J/g · °C. PP exponentially softens at around 140 °C. The thermal characteristics of PS measured by DSC, SDT, and TMA as well as PP are shown in Figure 2. The glass transition and sublimating temperatures are around 100 °C and 420 °C, respectively. The specific heat of PS is at 25 °C is 0.83 J/g · °C. PS shrinks in range from 100 °C to 130 °C and softens over 130 °C.

The Schematic diagram of the Cu cooling base is shown in Figure 3. The Cu plate with the thickness of 10 mm has four holes with a diameter of 1.2 mm. The polymer sheets are pressed to the base by extracting the air inside the gap between the sheet and the base through the four holes. The negative pressure is around 21 kPa.

The beam is focused on the surface of the sheets and scans 10 mm in length once with no assistant gas. Air is being extracting through the holes during laser operating. The effect of the cooling base is compared with a conventional method, which the backside of the sheet is air to prevent from being processed a stage which is usually used to support materials to be processed and not expected to be damaged.

## 3. Results and Discussion

### 3.1. Analytical Method

The center of the processed line, called groove in this literature, which is the center of the beam pointing position, is analyzed with a digital microscope (VHX-5000, KEYENCE, Osaka, Japan). The samples are coated with carbon using a carbon coater (CADE-E, Meiwafosis, Shinjuku, Japan) in order to accurately measure the shape of the grooves with the microscope. The coating thickness does not affect measurement since it is around 0.02 µm, which is much smaller than the sample thickness.

The three-dimensional shapes of the grooves of the PP sample processed at *S* = 100 mm/s and the PS sample at *S* = 150 mm/s are shown in Figure 4 and Figure 5, respectively. The picture sizes of the optical image are 1200 × 1600 pixels, the scale sizes are 1.11 µm/pixel, and the resolutions of the height are one micrometer. Figure 4c and Figure 5c show the height profiles at the dash line in the optical images in Figure 4b and Figure 5b. Depth used in this literature is defined by the distance between the initial surface of the sample and the bottom of the groove. Width is the groove width at the height of the initial surface. The errors are evaluated as ±3*σ* (*σ*: standard deviation). A wider range perpendicular to the laser-scanning direction than 1200 pixels is selected to measure the overall HAZ and initial height of the surface if the image has not enough size to measure the grooves.

### 3.2. Thin-Film Processing of PP

Figure 6 shows the depth and width of the PP samples with or without the Cu base. The horizontal axes are standardized to represent energy per unit volume, *E*_v_ = *P*/(*S*·*A*), where *A* is the focused area of the laser beam. In the case of air, the depth increases as *E*_v_ does up to ~2.5 × 10^10^ J/m^3^. Excessing the energy, the samples are partially penetrated without becoming thinner than 100 µm, which is the reason why there are two marks at the same *E*_v_ in Figure 6a. Finally, they are completely cut over *E*_v_ ~ 3.3 × 10^10^ J/m^3^. It indicates that it is quite difficult to make PP sheets thinner than the thickness of 100 µm with the conventional method. In the case of the Cu base, though the depth is a little shallower than that of air at the same laser condition, the samples are not cut even if *E*_v_ excesses the energy that the samples processed without the base are entirely cut. Increase of the depth stops at the value of around 250 µm at *E*_v_ ~ 3.3 × 10^10^ J/m^3^. The PP sheets are processed stably to around 250 µm in thickness up to *E*_v_ = 10 × 10^10^ J/m^3^. The error of the constant thin-film processing is ±20 µm. The depth starts decreasing over 10 × 10^10^ J/m^3^. The width becomes wider as the energy increases in both cases. The width in the case of the Cu base is wider than that of air. The blue dot-line in Figure 6a represents a linear approximation to the results of the Cu base in the range of the monotonic increase. The approximate equation, which is estimated by the method of least squares, is $D=74E_{v}-31$ where *D* is depth. The ablation threshold is found to be 4.7 kJ/g by using the PP density of 0.9 g/cm^3^ [21], which is comparable with that of other polymers [27,28].

Figure 7 shows the height profiles of the PP samples at (**a**) *E*_v_ = 3.8 × 10^10^ J/m^3^, (**b**) *E*_v_ = 9.5 × 10^10^ J/m^3^, (**c**) *E*_v_ = 38 × 10^10^ J/m^3^. The sample at *E*_v_ = 3.8 × 10^10^ J/m^3^ is sharply processed. The higher the laser energy is, the wider width and HAZ become.

The PP sheets with the Cu cooling base are never cut no matter how the energy is applied to them. It is considered that the cooling effect of the base on the sheet exceeds heating the sheet by the laser beam; that is, the boundary between the sheet and the base is in thermal equilibrium. Cu has high heat conductivity that is around 400 W/m·K, which is much higher than that of PP, 0.1–0.2 W/m·K [29]. In this experiment, heat transfer coefficient is also an important parameter. However, it is difficult to measure it during processing since the laser scanning speed is fast and the area heated by the laser beam is narrow. Gas generated by PP sublimating is blowing from the point heated by the laser beam and no flame appears during processing, known as laser ablation [28]. It is known that many types of hydrocarbons are generated by thermal decomposition of PP [30]. The products include approximately ten percentages of propylene in which the central wavelength for the C-H vibration mode is 10.684 µm [31]. Propylene gas absorbs a CO_2_ laser beam since the wavelength is quite near to that of the beam. Pressure pressing PP to the base increases by volume expansion by gasification of PP and heating propylene gas. The width of HAZ shown on both sides of the groove in Figure 4c is around 300 µm. It is considered that the PP sheet is heated to the backside in excess of the melting temperature since the HAZ width is comparable with the thickness of the PP sheet. In injection molding, liquified PP is compressed to a mold under high pressure and cooled down by the mold. Heat transfer coefficient from liquid PP to the mold in injection molding, which steel is preferred to be used as a mold because of the durability [32], is approximately 2.5 kW/m^2^·K [33]. The value is comparable to water forced convection. It is supposed that the backside of the liquified PP during laser processing is being cooled down by being pressed to the cooling base. It is difficult to explain why the depth decrease despite increase of the energy applied to it. The thickness of the sample processed at the laser condition of *E*_v_ = 9.5 × 10^10^ J/m^3^, is evaluated with a laser interferometer (SI-F1000V and SI-F80, KEYENCE, Osaka, Japan). The thicknesses at 2 mm, 5 mm (center), and 8 mm from the initial point that the laser starts scanning are around 40 µm, 60 µm, and 70 µm, respectively. The thickness increases as approaches to the initial point. The generated gas blows faster in the horizontal direction, parallel to the groove, as the energy increases. The points nearer to the initial laser point are more affected by the gas. The reason of the thickness increase might be deformation and flattening by the gas heating the groove and dross adhesion.

The reason why the width of the Cu case is wider than that of the air case is considered as below. The transmittance of the PP sheets is 40% and reflectivity of Cu at the wavelength of 10.6 µm is around 99% [34]. The sheets, therefore, is heated by the reflected beam from the Cu base. 

### 3.3. Thin-Film Processing of PS

Figure 8 shows the depth and width of the PS samples with or without the Cu base. In the air case the thinner the samples become, the deeper the depth gets. The samples start penetrating partially and totally cut at *E*_v_ ~ 3.3 × 10^10^ J/m^3^. In the case of the Cu base, the depth increases as the energy does and the samples are totally cut after partial penetration as well as in the air case. The increasing ratio of the widths in both cases are the same at the same *E*_v_ unlike PP. 

The height profile of the PS sample at *E*_v_ = 10 × 10^10^ J/m^3^ is shown in Figure 5c. Compared with PP, the shape of the groove is gentle, the height of HAZ is high, and the width is narrow.

The sharp increase of the depth in the air case over *E*_v_ ~ 0.8 × 10^10^ J/m^3^ results from expansion of the backside. As the samples thin, the backside enlarges because of heat and pressure. It makes quality of products worse and thin-film processing of PS difficult. The depth in the case of the Cu base seems to increase at the same rate of the air case in *E*_v_ < 0.8 × 10^10^ J/m^3^. It is supposed that expansion of the backside is suppressed since the backside of the sample is supported and cooled by the base. It is considered that the base cools the PS sheet the same degree as the effects on PP because main component of the gas generated by decomposition of PS is styrene [35], which has strong absorption peak at 10.6 µm [36], and PS has the same cooling curve in molding as PP reported in [37]. Thin-film processing of PS sheets is realized at *E*_v_ ~ 2.5 × 10^10^ J/m^3^ with error of ±40 µm. The roughness at the groove bottom is worse than that of PP. Since PS shrinks at glass transition temperature before softening as shown in Figure 2c, PS continues to deform by residual heat after the laser scanning. The center point in the area heated by the laser beam has higher temperature than that of both sides because of intensity distribution of the Gaussian beam. When the center temperature is beyond the melting temperature and the side ones are at the glass transition temperature, the center can stretch and is pulled by the side shrinking. The uncontrollable phenomenon results in expansion of the grooves in the cooling process of PS and rougher quality of the groove bottom. The blue dot-line in Figure 8a represents a linear approximation to the results of the Cu base in the range with no penetration. The approximate equation, which is estimated by the method of least squares, is $D=98E_{v}+3.7$. It is difficult to find the ablation threshold in this way because the depth includes not only the amount of thermal processing but also deformation by residual heat as described above. 

The reason why the widths of both cases increase at the same rate is considered that PS is not reheated by the reflected beam from the base since PS has higher absorptivity than PP.

### 3.4. Characteristics Requierd to the Cooling Base

At first, materials of the base have to have high heat conductivity in order to effectively let heat from polymers away. The base also needs high heat capacity, which is volume multiplied by specific heat. Temperature rise of the base should be ignorable. 

Secondly, the base should not absorb CO_2_ laser beams. Absorption of the laser beam causes the base temperature rising, heating polymer sheets by the base, and damaging the base. Damage or deformation of the base make the quality of products worse. The durability of the base is desired to be long enough industrially and economically. Cu does not absorb but reflects a CO_2_ laser beam since it has high reflectivity at the wavelength of 10.6 µm. If the base reflects the beam like Cu, the sheets are heated by the reflected beam from the base. 

Thirdly, the base has to be stable at the polymers sublimating temperature in order to prevent it from damage or deformation. Most metals have higher melting temperature than the sublimating temperature.

### 3.5. Other Metals 

Materials which have similar characteristic to Cu as described in the last section should be available as the cooling base. Al, brass, and steel are used as the cooling base. Their optical and thermal characteristics are shown in Table 1. Al and brass have little absorptivity and relatively high thermal conductivity as well as Cu, while steel (SS400 in Japanese Industrial Standards (JIS)) has a little higher absorptivity and lower thermal conductivity than Cu. All bases have the same size and structure. This experiment is performed with the PP sheets under the same condition as described in Section 2.

Comparison of the thickness of the PP samples in the case of the Cu base with that of Al, brass, and steel is shown in Figure 9. The thicknesses are measured with the laser interferometer used in Section 3.1. The width and error bars in both figures are evaluated with the digital microscope as well as the above-mentioned method. The thickness is approximately 50 ± 20 µm in each base.

The height profiles of the grooves processed with each metal at *E*_v_ = 5.1 × 10^10^ J/m^3^ are shown in Figure 10. There is no great difference in shape among the metals.

It is confirmed that the four materials can be used as the cooling base to thin the polymer sheets. The results indicate that materials which have low absorption of the wavelength of 10.6 µm, high thermal conductivity, and higher melting temperature than the sublimating temperature of polymers are available to the cooling base.

## 4. Conclusions

We developed the new method of thin-film processing of the PP and PS sheets by cooling the backside of the sheets by the Cu base. The method can process the sheets to around 50 µm in thickness. The PP sheets have never cut with the method no matter how the amount of laser energy per unit volume increases, while the width of the grooves gets wider as the energy increases. The increase of the groove depth stops at *E*_v_ ~ 3.3 × 10^10^ J/m^3^. It is considered that the boundary between the backside of the sheet and the cooling base is in thermal equilibrium because of high heat conductivity of the base. The PS sheets can be processed to thin thickness with less deformation of the backside owing to the support of the base, compared with the conventional method that does not use the stage under the sheets in order not to damage it. The PS sheets have completely cut at *E*_v_ > 3.2 × 10^10^ J/m^3^. The difference between PP and PS results from the thermal–mechanical characteristic of PS, which shrinks at the glass transition temperature. It is considered that other metals of which optical and thermal characteristics are similar to those of Cu can be available to the base. The characteristics are low absorptivity at the wavelength of 10.6 µm, high heat conductivity, and higher melting temperature than the sublimating temperature of the polymers. It is confirmed that aluminum, brass, and steel can be used as the base as well as Cu though steel has higher absorptivity and lower heat conductivity than those of Cu. The advantages of the method are that the metals are obtainable and not so expensive, it is easy to install to existing equipment with a minor change of the stage, and complex controls of the system are not needed.

## Figures and Tables

**Figure 1 polymers-13-01448-f001:**
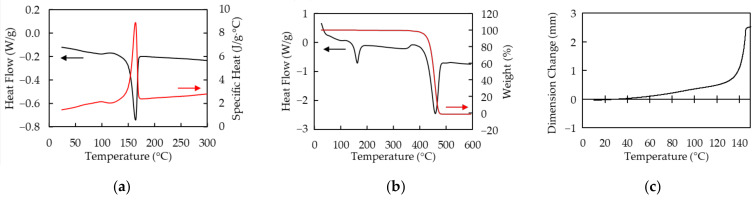
The thermal characteristics of polypropylene (PP) are shown. All measurements are performed under nitrogen atmosphere. (**a**) Heat flow (black line) and specific heat (red line) measured by differential scanning calorimetry (DSC); (**b**) Heat flow (black line) and weight (red line) measured by simultaneous DSC and thermogravimetric analysis (SDT) under nitrogen atmosphere; (**c**) Dimensional change measured by thermal–mechanical analysis (TMA). Static pull force of 0.1 N is applied to a sample.

**Figure 2 polymers-13-01448-f002:**
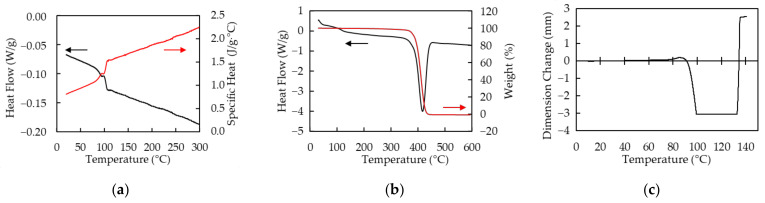
The thermal characteristics of polystyrene (PS) are shown. All measurements are performed under nitrogen atmosphere. (**a**) Heat flow (black line) and specific heat (red line) measured by DSC; (**b**) Heat flow (black line) and weight (red line) measured by SDT under nitrogen atmosphere; (**c**) Dimensional change measured by TMA. Static pull force of 0.1 N is applied to a sample.

**Figure 3 polymers-13-01448-f003:**
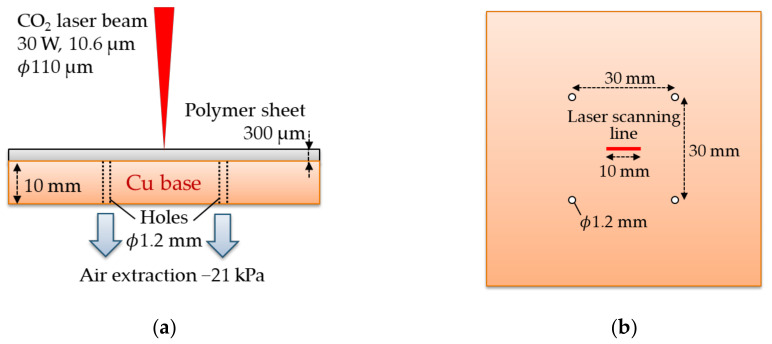
Side (**a**) and top (**b**) views of the Cu cooling base. The base has four holes with a diameter of 1.2 mm. The sheet is pressed to the base with negative pressure of 21 kPa.

**Figure 4 polymers-13-01448-f004:**
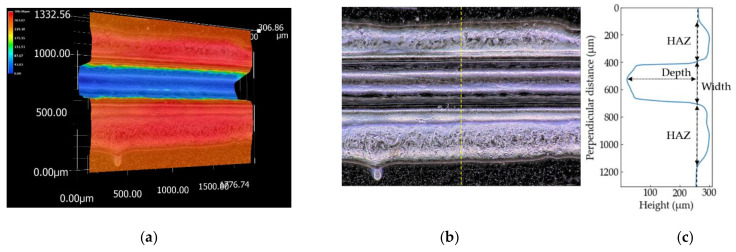
Three-dimensional shape (**a**), optical image (**b**) and the height profile at the yellow dash line (**c**) of the PP sample measured by the digital microscope. The sample is processed at *S* = 100 mm/s.

**Figure 5 polymers-13-01448-f005:**
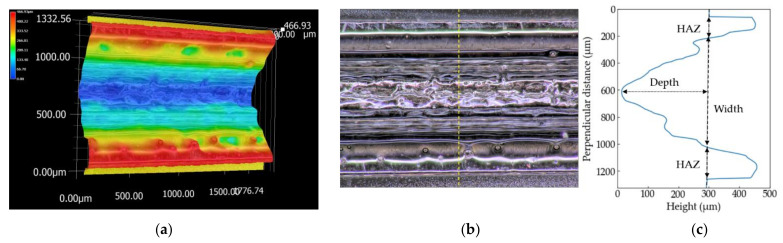
Three-dimensional shape (**a**), optical image (**b**) and the height profile at the yellow dash line (**c**) of the PS sample measured by the digital microscope. The sample is processed at *S* = 150 mm/s.

**Figure 6 polymers-13-01448-f006:**
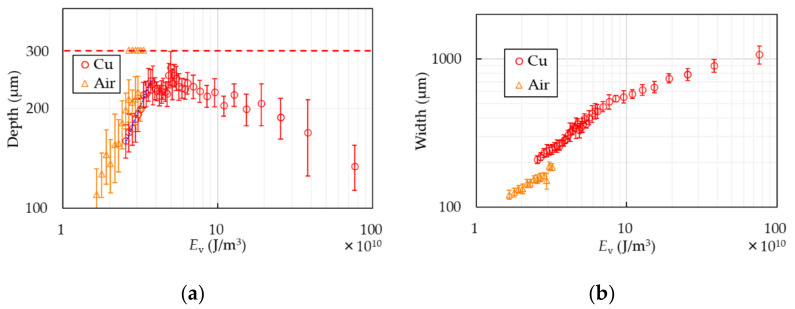
Depth (**a**) and width (**b**) of PP are shown. The circles show the results of the Cu cooling base and the triangles show that of air. The dash line in (**a**) represents the sample thickness. Two triangles at the same *E*_v_ in (**a**) mean partially penetration. The blue dot-line in (**a**) represents linear approximation to the results of the Cu base in the range of the monotonic increase.

**Figure 7 polymers-13-01448-f007:**
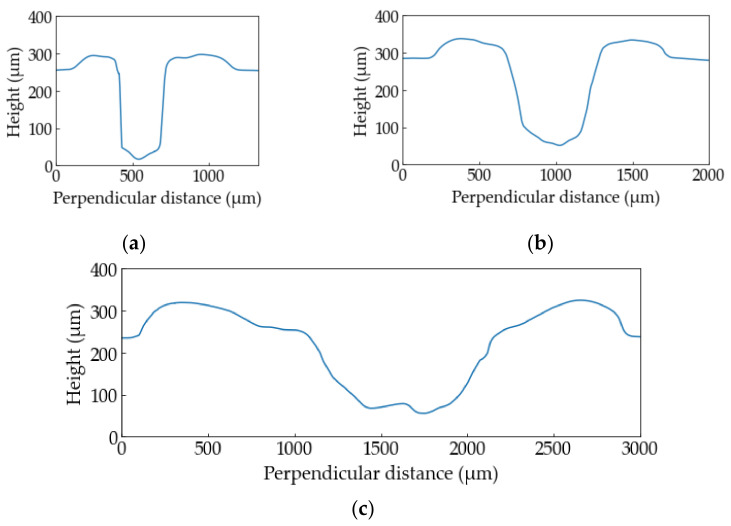
The height profiles of the PP samples: (**a**) *E*_v_ = 3.8 × 10^10^ J/m^3^, (**b**) *E*_v_ = 9.5 × 10^10^ J/m^3^, (**c**) *E*_v_ = 38 × 10^10^ J/m^3^.

**Figure 8 polymers-13-01448-f008:**
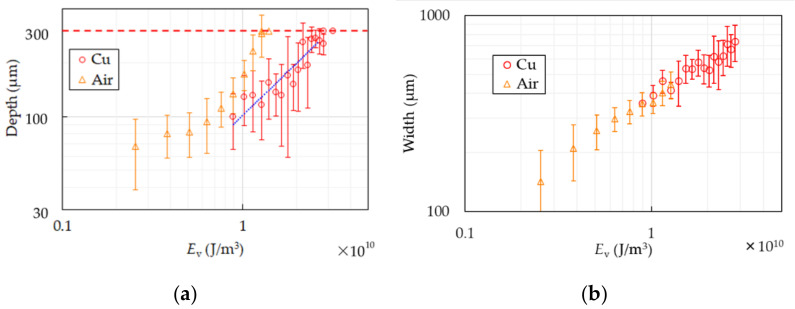
Depth (**a**) and width (**b**) of PS are shown. The circles show the results of the Cu cooling base and the triangles show that of air. The dash line in (**a**) represents the sample thickness. Two circles and triangles at the same *E*_v_ in (**a**) mean partially penetration. Single circle and triangle with no error bar means complete cutting. The blue dot-line in (**a**) represents linear approximation.

**Figure 9 polymers-13-01448-f009:**
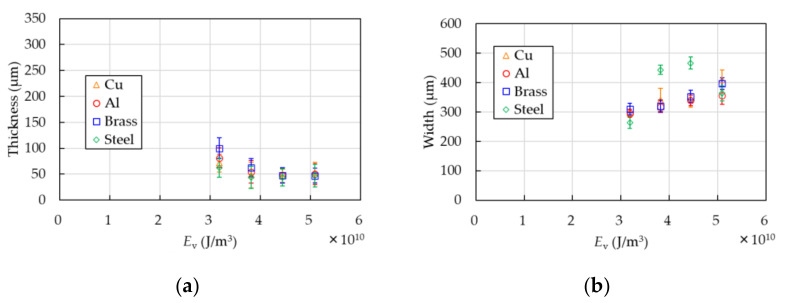
Comparison of the Cu base with that of Al, brass, and steel. Thickness (**a**) and width (**b**) are shown. The thickness is measured with the laser interferometer. The width and error bars in both figures are evaluated with the digital micrometer as well as the above-mentioned method.

**Figure 10 polymers-13-01448-f010:**
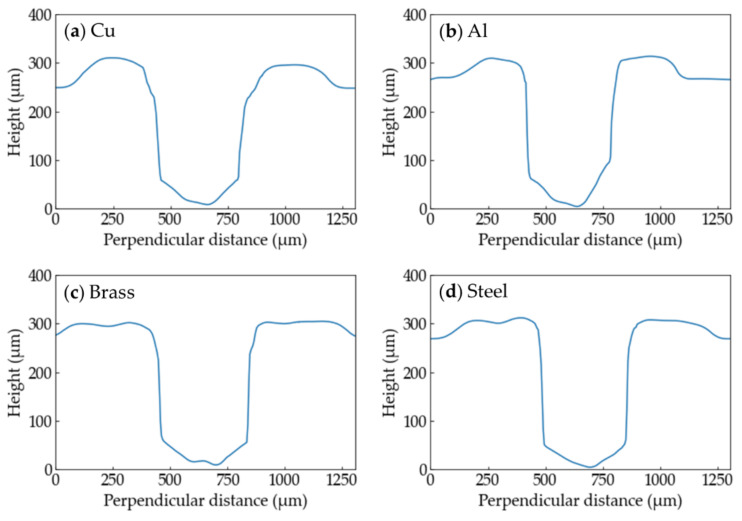
Height profiles of the grooves of the PP samples processed with each metal at *E*_v_ = 5.1 × 10^10^ J/m^3^ are shown. (**a**) Cu; (**b**) Al; (**c**) brass; (**d**) Steel.

**Table 1 polymers-13-01448-t001:** Optical and thermal characteristics of the metals used as the cooling base.

Metal	Absorption of 10.6 µm (%)	Thermal Conductivity (W/m·K)	Melting Temperature (°C)
Cu	~1 [34]	401 @ 0 °C	1084
Al	~1 [34]	236 @ 0 °C	660
Brass	~2 [38]	111 @ 20 °C	930
Steel	~10 ^1^ [39]	~50 @ 20 °C ^2^	~1500 ^2^

^1^ The value of steel is shown. ^2^ The values of steel carbon are shown.

## Data Availability

The data presented in this study are available on request from the corresponding author.

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
