# Peer review of "Thin-Film Processing of Polypropylene and Polystyrene Sheets by a Continuous Wave CO_2_ Laser with the Cu Cooling Base"

_polymers, 2021, doi:10.3390/polym13091448_

Round 1
Reviewer 1 Report
It needs to be revised. See other manuscripts and compare with your results. How innovative are your results? Have you done similar research before? Can't see the clear goal of the research? The conclusion is not clear.
Reviewer 2 Report
- In introduction it should be explained what for (what kind of practical application) is that kind of half-cutting technique
- I believe, absorption coefficient of PP and PS should be provided (as determined or referenced from literature) - the difference in how much energy can be converted into heat is crucial!
- Ablation characteristic (at least ablation threshold) for each of polymer films should be determined and the mechanisms for the laser cutting explained.
- For fig,. 10 it is not clear to what polymers (PP, PS) it refers to.
- In my opinion conclusion are too general and more detailed conclusions would provide better understanding of scientific outcomes.
